RESEARCH                                                                    **Open Access**

# Chromatin regulates expression of small RNAs to help maintain transposon methylome homeostasis in Arabidopsis

Ranjith K. Papareddy, Katalin Páldi[†], Subramanian Paulraj[†], Ping Kao, Stefan Lutzmayer and Michael D. Nodine[*]

* Correspondence: michael.nodine@gmi.oeaw.ac.at
[†]Katalin Páldi and Subramanian Paulraj contributed equally to this work.
Gregor Mendel Institute (GMI), Austrian Academy of Sciences, Vienna Biocenter (VBC), Dr. Bohr-Gasse 3, 1030 Vienna, Austria

## Abstract

**Background:** Eukaryotic genomes are partitioned into euchromatic and heterochromatic domains to regulate gene expression and other fundamental cellular processes. However, chromatin is dynamic during growth and development and must be properly re-established after its decondensation. Small interfering RNAs (siRNAs) promote heterochromatin formation, but little is known about how chromatin regulates siRNA expression.

**Results:** We demonstrate that thousands of transposable elements (TEs) produce exceptionally high levels of siRNAs in *Arabidopsis thaliana* embryos. TEs generate siRNAs throughout embryogenesis according to two distinct patterns depending on whether they are located in euchromatic or heterochromatic regions of the genome. siRNA precursors are transcribed in embryos, and siRNAs are required to direct the re-establishment of DNA methylation on TEs from which they are derived in the new generation. Decondensed chromatin also permits the production of 24-nt siRNAs from heterochromatic TEs during post-embryogenesis, and siRNA production from bipartite-classified TEs is controlled by their chromatin states.

**Conclusions:** Decondensation of heterochromatin in response to developmental, and perhaps environmental, cues promotes the transcription and function of siRNAs in plants. Our results indicate that chromatin-mediated siRNA transcription provides a cell-autonomous homeostatic control mechanism to help reconstitute pre-existing chromatin states during growth and development including those that ensure silencing of TEs in the future germ line.

**Keywords:** Small RNAs, DNA methylation, Chromatin, Epigenetics, Linker histone H1, Plant embryogenesis, RNAi, Transposable elements

## Background

Eukaryotic genomes are partitioned into euchromatic and heterochromatic domains [1, 2]. Euchromatic regions are enriched for genes and provide a transcriptionally permissive state. Heterochromatic regions are densely packed, or condensed, regions of the genome that are typically transcriptionally quiescent and characterized by highly

repetitive DNA and transposable elements [3, 4]. Heterochromatin formation is promoted by various pathways including those affecting covalent modifications of histones, which package DNA into nucleosomes, as well as cytosine methylation [5–7]. Chromatin states are re-established after fertilization of egg and sperm in diverse animals [8–12], and histone reprogramming has also been observed in plants shortly after fertilization [13, 14]. However, little is known about how heterochromatin-promoting pathways respond to labile chromatin states shortly after fertilization to help re-establish euchromatic and heterochromatic states.

Small RNA-based pathways promote heterochromatin formation and associated transcriptional silencing in animals, fungi, and plants [15]. Plants employ 24-nucleotide (nt) small interfering RNAs (siRNAs) to help promote silencing of repetitive elements including TEs, and thus prevent DNA mutations caused by TE mobilization. During canonical RNA-directed DNA methylation (RdDM), RNA Polymerase IV (Pol IV) is recruited to target loci and generates transcripts that are co-transcriptionally converted into double-stranded RNAs by RNA-DEPENDENT RNA POLYMERASE 2 (RDR2) [16–21]. The resulting double-stranded RNAs are then processed into 23-nt/24-nt RNA duplexes by the DICER-LIKE 3 (DCL3) endoribonuclease [22]. The 24-nt strand of the duplex binds to ARGONAUTE 4 (AGO4) and guides AGO4 and associated proteins to siRNA-complementary sites contained within noncoding RNAs produced by RNA Polymerase V [23, 24]. DOMAINS REARRANGED METHYLTRANSFERASES 1/2 (DRM1/2) is then recruited to target loci, which results in the de novo methylation of cytosines in the CG, CHG, and CHH contexts (where H ≠ G) [25, 26]. Because CG and CHG methylation are maintained independently of siRNAs by DNA METHYLTRANSFERASE 1 (MET1) and CHROMOMETHYLASE 3 (CMT3), respectively, CHH methylation is a good indicator of RdDM activities [20]. Nevertheless, CHH methylation can also be maintained independently of siRNAs by CMT2 in post-embryonic tissues, and this occurs mostly on the bodies of long TEs that are densely packed in nucleosomes and inaccessible to DRM2 [27].

Although sRNAs promote heterochromatin formation in diverse eukaryotic species, little is known about how chromatin states regulate small RNA production in animals and plants. Plant gametes and associated companion cells that support the gametes are contained within multicellular haploid gametophytes [28]. Current models propose that the large-scale chromatin decondensation observed in terminally differentiated companion cells facilitates TE transcription, and the resulting transcripts serve as substrates for RDR1/6 and DCLs to generate 20–22-nt siRNAs that move into gametes and stabilize TE silencing [29–33]. The endosperm and embryo are products of double-fertilization, and it has also been suggested that hypomethylation of endosperm promotes the production of siRNAs, which then move into the embryo to mediate TE methylation [31, 34]. RdDM is required for the progressive methylation of a few target loci during *Arabidopsis thaliana* (Arabidopsis) embryogenesis [35], and late-staged embryos are CHH hyper-methylated compared to other tissues [36–38]. However, the dynamics of embryonic RdDM have not been reported genome-wide in plants due to the difficulty in generating genome-wide profiles of siRNAs and methylomes from early embryos, which are small and deeply embedded within maternal seed tissues. More generally, it is virtually unknown how chromatin states may promote the de novo production of

sRNAs to help re-establish TE silencing cell-autonomously including in the future germ-line.

## Results

### Embryos are enriched for transposon-derived small RNAs

We recently developed a low-input small RNA sequencing (sRNA-seq) method to profile small RNA dynamics during eight stages of Arabidopsis embryogenesis, as well as floral buds and leaves [39] (Fig. 1a and Additional file 1: Table S1). We previously

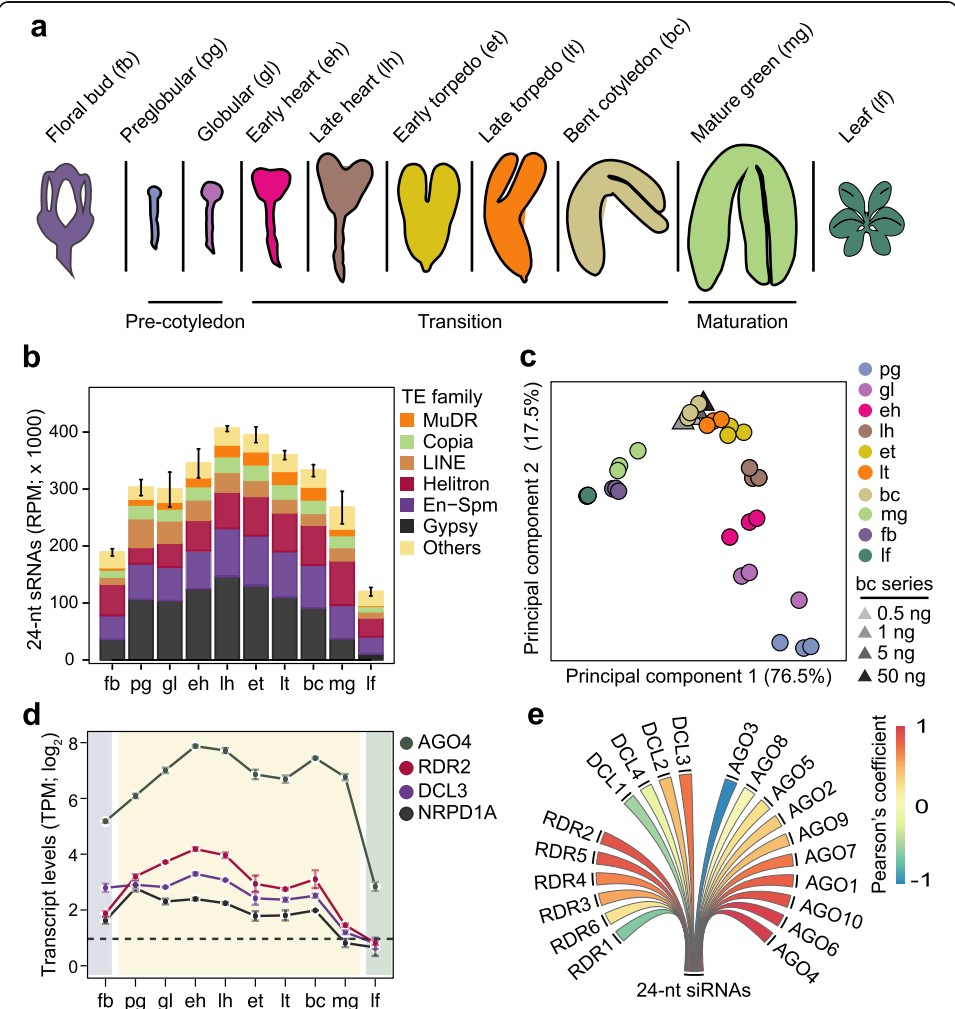

**Fig. 1** Embryos are enriched for transposon-derived small RNAs. **a** Schematic of embryonic stages and tissues previously used for small RNA profiling [39] and analyzed in this study. **b** Stacked bar chart representing the abundance of 24-nt sRNAs from various TE families in floral buds, embryos, and leaves (key). Error bars indicate standard error of mean TE-derived 24-nt siRNA levels from three biological replicates. RPM, sRNA-seq reads per million genome-matching reads. **c** Principal component analysis of 24-nt sRNAs mapping to TEs in floral buds, embryos, leaves, and a dilution series of RNA isolated from bent cotyledon stage embryos (key). **d** Line chart illustrating transcript levels of 24-nt siRNA biogenesis factors in embryonic and post-embryonic tissues based on mRNA-seq [40]. Error bars represent standard error of mean transcripts from three biological replicates. TPM, transcripts per million. **e** Pearson's correlation coefficients between means of TE-derived 24-nt siRNAs and DCL, RDR, and AGO transcript levels in floral buds, embryos, and leaves. Pearson's correlation coefficient values are represented according to the key. See also Additional file 2: Figure S1

focused on the ~ 21-nt microRNA class of small RNAs involved in post-transcriptional regulation [39], but noticed that the vast majority of TE-derived small RNAs were 24-nt long and highly enriched in embryos compared to floral bud or leaf tissues (Fig. 1b and Additional file 2: Figure S1A). Small RNAs were detected from 28,087 TEs and the highest amounts from several families including Gypsy, MuDR, and En-Spm were detected during mid-embryogenesis. The levels of TE-derived 24-nt sRNAs were highly correlated among biological replicates from floral bud, embryonic, and leaf tissues indicating stage- and tissue-specific sRNA populations (Fig. 1c and Additional file 2: Figure S1B). Moreover, principal component analysis revealed that 76.5% and 17.5% of the variation in TE-derived 24-nt siRNAs was accounted for by principal components 1 and 2, respectively. Principal component 1 distinctly separated post-embryonic and mature green embryo stages from pre-maturation embryonic stages, whereas principal component 2 stratified the pre-maturation embryonic samples according to developmental stage. Libraries prepared from 50, 5, 1, or 0.5 ng of RNA isolated from bent cotyledon embryos clustered together with the biological replicates generated from ≥ 500 ng of bent cotyledon RNA (Fig. 1c and Additional file 2: Figure S1B). This indicated that the vast majority of variation observed in 24-nt embryonic and post-embryonic siRNA populations was biological rather than technical.

Small interfering RNAs involved in RdDM are typically 24-nt long and begin with a 5′ adenine [41]. Accordingly, adenosines were the dominant first base of 24-nt sRNAs in embryos (Additional file 2: Figure S1C), and the levels of embryonic 24-nt sRNAs were most highly correlated with the levels of transcripts encoding key canonical RdDM components such as RDR2, DCL3, and AGO4, which were also enriched in developing embryos (Fig. 1d, e) [20, 40]. Therefore, canonical 24-nt siRNAs are highly enriched in embryos, exhibit distinct developmental dynamics, and upon maturation become similar to post-embryonic siRNA populations.

### Small RNAs from euchromatic and heterochromatic transposons exhibit distinct developmental dynamics

To examine the temporal dynamics of embryonic siRNAs mapping to TEs in more detail, we used *mclust* [42] to define eight unique clusters of the 31,189 TAIR10-annotated TEs based on their 24-nt siRNA levels (Additional file 2: Figure S2A, B). At least 2 sRNA-seq reads per million genome-matching reads were detected for 11,845 TEs, and these were grouped into either class A (6116 TEs; 19.6% of total) or class B (5729 TEs, 18.4% of total) based on the dynamics of their corresponding siRNAs between floral buds, developing embryos, and leaves (Fig. 2a, b, Additional file 3: Table S3, and Additional file 2: Figure S2B). The remaining 19,344 TEs (62.0% of total) were considered siRNA-depleted (Fig. 2a and Additional file 2: Figure S2B, C) and served as negative controls. Small interfering RNAs from class A TEs had low levels in preglobular embryos that were increased at the globular stage, and were stable until sharply increasing at the mature green stage and then remained at high levels in leaves and floral buds (Fig. 2a–c). In contrast, class B TEs produced large amounts of siRNAs already in preglobular embryos, then continued to gradually increase through mid-embryogenesis and were strongly reduced in mature embryos, leaves, and floral buds (Fig. 2a–c).

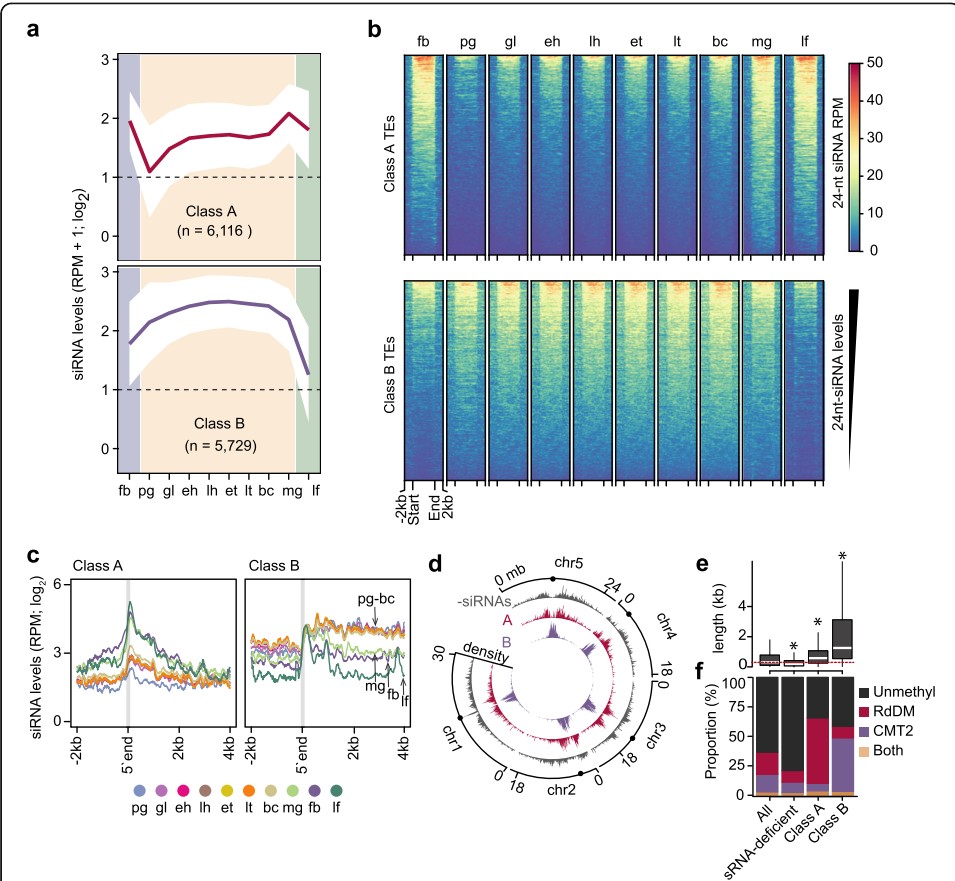

**Fig. 2** Small RNAs from euchromatic and heterochromatic transposons exhibit distinct developmental dynamics. **a** Line graphs illustrating 24-nt siRNA levels from class A and B TEs in embryonic and post-embryonic tissues. Dashed lines represent the detection criteria used to select TEs yielding siRNAs (2 RPM, reads per million genome-matching reads). The number of TEs belonging to each class are indicated. Polygons represent the standard deviation of mean 24-nt siRNA levels. fb, floral buds; pg, preglobular; gl, globular; eh, early heart; lh, late heart; et, early torpedo; lt, late torpedo; bc, bent cotyledon; mg, mature green; lf, leaf. **b** Heat map depicting 24-nt siRNA levels (RPM) from class A and B TEs across development. siRNA levels from ± 2 kb of TEs are color-coded according to the key, and samples are labeled as in **a**. Rows were ordered based on total 24-nt siRNA levels. **c** Line metaplot showing 24-nt siRNA levels from class A and B TEs. TEs were aligned on their 5′ ends, which are indicated by the gray vertical line. Samples are labeled according to the key and as in **a**. **d** Circos plot representing densities of siRNA-deficient (-siRNAs; gray), class A (red) and class B (purple) TEs across the five Arabidopsis chromosomes (chr). Centromeres are indicated by dots. **e** Boxplot of TE lengths for either all annotated, siRNA-deficient, class A, or class B TEs. Dashed red line represents the median size of all annotated TEs. Thick horizontal bars indicate medians, and the top and bottom edges of the box indicate the 75th and 25th percentiles, respectively. kb, kilobases; *P* values < 0.0001 based on Mann-Whitney *U* test of differences between all annotated and TE classes are represented by *. **f** Stacked bar charts illustrating proportion of TEs that are CHH hypomethylated in *drm1/drm2* (RdDM; red), *cmt2* (CMT2; purple), both *drm1/drm2* and *cmt2* (both; yellow) or which were not methylated (unmethyl; black) in leaves [26]. See also Additional file 2: Figure S2

Class A and B TEs have distinct features. Class A TEs are short and dispersed along pericentromeric and euchromatic regions of chromosomes (Fig. 2d, e and Additional file 2: Figure S2D). In contrast, class B TEs are generally longer, concentrated in heterochromatic centromeres, and especially in embryos, siRNAs are generated from throughout whole TEs (Fig. 2c–e and Additional file 2: Figure S2D). Transposons targeted by DRM2 and CMT2 are distinct to euchromatic and heterochromatic domains of the genome, respectively [27]. The sizes and genomic locations of class A and B TEs are

characteristic of TEs respectively methylated by either the RdDM or CMT2 pathways in post-embryonic tissues [26, 27, 43]. Indeed, class A TEs were enriched for TEs with reduced methylation in RdDM-defective *drm1/2* mutant leaves including HAT, SINE, SADHU, and other short TE families, whereas class B TEs were enriched for TEs with reduced methylation in *cmt2* mutant leaves including MuDR, En-Spm, and long terminal repeat families such as Gypsy and Copia (Fig. 2f and Additional file 2: Figure S2E, F). In addition, class B TEs have heterochromatic features such as high levels of GC content, nucleosome occupancy, HISTONE 3 LYSINE 9 di-methylation (H3K9me2), and linker histone 1 (H1) compared to class A TEs (Additional file 2: Figure S2G). Altogether, we identified two distinct classes of TEs based on the levels of siRNAs they produce during development: euchromatic TEs (i.e., class A) that progressively generate siRNAs during embryogenesis and are methylated by the siRNA-dependent RdDM pathway in post-embryonic tissues, and heterochromatic TEs (i.e., class B) that produce very large amounts of siRNAs during embryogenesis prior to maturation and are methylated independently of siRNAs in post-embryonic tissues.

### Embryonic methylome dynamics

During RdDM, 24-nt siRNAs are loaded onto ARGONAUTE proteins and serve as sequence-specific guides for the recruitment of methyltransferases to target loci. siRNA-directed methylation of TEs contributes to their transcriptional silencing and immobilization and limits their mutagenic potential [44–47]. To investigate the functions of embryonic 24-nt siRNAs, we adapted a whole-genome bisulfite sequencing approach called methylC-seq [48] to profile methylomes at single-base resolution from the low amounts of DNA available from early Arabidopsis embryos (see the "Methods" section) (Additional file 1: Table S1). Comparisons of methylomes generated with 0.1, 0.5, 1, or ~ 4 ng of genomic DNA isolated from bent cotyledon embryos had nearly identical cytosine methylation levels indicating that there was low variability of this method when using different amounts of input DNA (Additional file 2: Figure S3A). Therefore, we used this robust low-input methylC-seq method to profile methylomes from 8-cell/16-cell (preglobular; 3 days after pollination [DAP]), early heart (4 DAP) and bent-cotyledon (8 DAP) embryos, as well as leaves and floral buds. We compared these datasets with publicly available methylomes generated from sperm [32], or late-staged embryos from early torpedo [49], mid-torpedo to early maturation [31], or mature green [37] stages. Because methylation of cytosines in the CHH context (mCHH, where H ≠ G) is a hallmark of siRNA-directed DNA methylation [20], we focused on CHH methylation. More specifically, the Arabidopsis genome was divided into 50-kb bins, and the mean-weighted CHH methylation rates from reproductive, embryonic, and vegetative tissues were calculated for each bin (Additional file 2: Figure S3B). Consistent with previous studies, CHH methylation gradually increased during embryogenesis until peaking at the maturation stage and then decreased in leaves (Additional file 2: Figure S3B) [35, 37, 38]. As expected, CHH methylation was most prominent at pericentromeric and centromeric regions densely populated with euchromatic and heterochromatic TEs that were enriched for siRNAs during embryogenesis (Additional file 2: Figure S3B). Relative to sperm, euchromatic TEs had low CHH methylation levels in early embryos that increased during embryogenesis and were hypermethylated relative

to leaves in 8 DAP bent-cotyledon-staged embryos (Fig. 3a). In contrast, CHH methylation was barely detectable from heterochromatic TEs in sperm, but then increased in early embryos and became hypermethylated relative to leaves by 6 DAP in early torpedo-staged embryos (Fig. 3b). Therefore, CHH methylation is established on both euchromatic and heterochromatic TEs during early embryogenesis, and TEs become hypermethylated relative to post-embryonic tissues at late stages of embryogenesis.

To examine embryonic methylation dynamics in more detail, we identified significant differentially methylated regions (DMRs) by pairwise comparisons between six embryonic stages (preglobular, early heart, early torpedo, bent cotyledon, late torpedo-to-early mature green and mature green) (see the "Methods" section). We found 21,361 embryonic CHH DMRs with a median size of approximately 100-bp (Additional file 4:

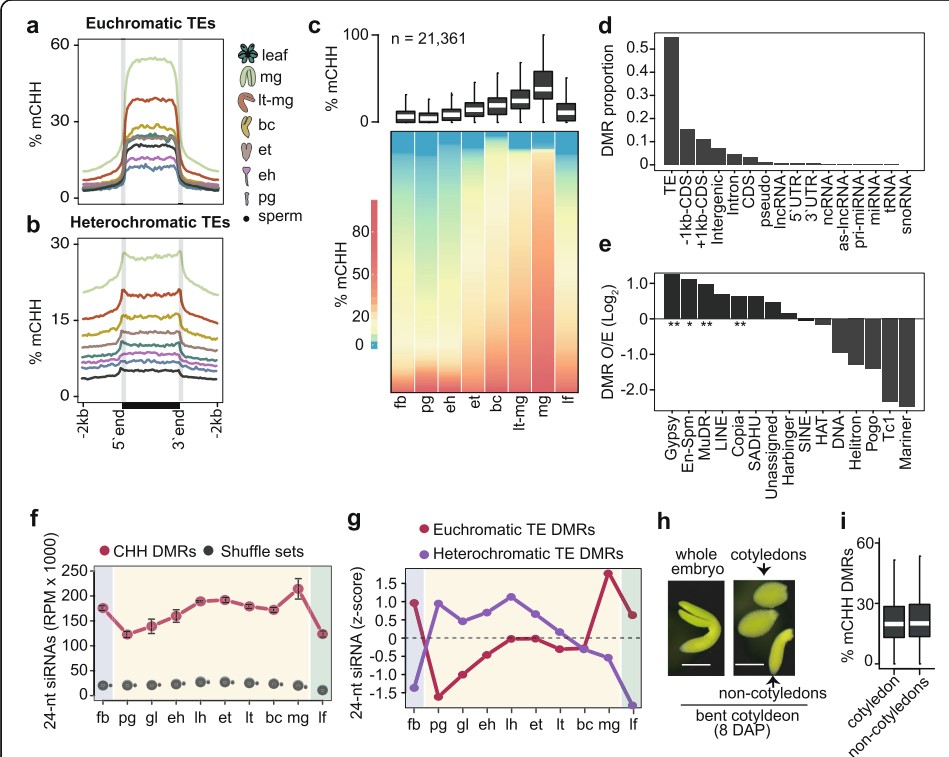

**Fig. 3** Embryonic methylome dynamics. **a**, **b** Metaplots of average weighted CHH methylation percentages across euchromatic (**a**) and heterochromatic (**b**) TEs in sperm, embryos, and leaves (key). pg, preglobular; eh, early heart; et, early torpedo; bc, bent cotyledon; lt-mg, late torpedo-to-early mature green; mg, mature green. **c** Boxplot and heat map illustrating the percentage of CHH methylation across DMRs in flowers, embryos, and leaves. The number of DMRs identified are indicated, and stages are labeled as in **a** and include floral buds (fb). Thick horizontal bars in the boxplot indicate medians, and the top and bottom edges of the box indicate the 75th and 25th percentiles, respectively. DMRs in heatmap were sorted by average methylation levels per column. **d** Proportion of genomic features overlapping DMRs. **e** Bar chart showing the enrichment of TE families observed overlapping DMRs relative to those expected based on their genomic proportions (O/E; log$_2$). *P* values < 0.01 and < 0.05 based on Fisher's exact test are represented by ** and *. **f** Line chart of 24-nt siRNA levels overlapping CHH DMRs (red) compared to randomly selected genomic regions with equal sizes and quantities (gray) in floral buds (fb), embryos, and leaves (lf). Embryo stages are labeled as in Fig. 1a. **g** Relative 24-nt siRNA levels (z-scores) on DMRs mapping to euchromatic (red) and heterochromatic (purple) TEs across development. Embryo stages are labeled as in Fig. 1a. **h** Representative image of a bent cotyledon-staged embryo 8 days after pollination (DAP) and dissected into cotyledon and non-cotyledon tissues for methylC-seq. Scale bar represents 0.25 mm. **i** Boxplot of CHH methylation percentages of DMRs defined in **c** for cotyledon and non-cotyledon tissues. Boxplots are as described in **b**. See also Additional file 2: Figure S3

Table S4, Additional file 2: Figure S3d). Consistent with the genome-wide CHH methylation dynamics described above, embryonic CHH DMRs were hypomethylated in preglobular embryos, progressively methylated until embryo maturation, and then were sharply reduced in leaves and floral buds (Fig. 3c and Additional file 2: Figure S4A). Moreover, these embryonic DMRs were observed on various TE families especially Gypsy and Copia LTR retrotransposons, as well as MuDR and En-Spm family DNA TEs (Fig. 3d, e). The enrichment of mCHH DMRs across various TE families was generally consistent with the corresponding levels of embryonic 24-nt siRNAs (Fig. 3e and Additional file 2: Figure S2F). To examine the relationships between siRNAs and methylation further, we quantified 24-nt siRNA levels on mCHH DMRs across embryogenesis. Compared to randomized controls, CHH DMRs were highly enriched for 24-nt siRNAs (Fig. 3f). siRNAs overlapping euchromatic TE DMRs were lowest in early embryos, peaked at maturation and were also abundant in floral buds and leaves. In contrast, siRNAs overlapping heterochromatic TE DMRs were highly abundant at early-to-middle stages of embryogenesis and then strongly reduced in late-embryonic stages, leaves, and floral buds (Fig. 3g). Therefore, large-scale changes of DNA methylation occur on both euchromatic and heterochromatic TEs during embryogenesis and are associated with 24-nt siRNAs.

Notably, both euchromatic and heterochromatic TEs were hypermethylated in mature embryos relative to earlier stages and post-embryonic tissues (Fig. 3a–c and Additional file 2: Figure S3B). Because the proportion of embryonic tissue composed of cotyledons also increases during embryo development, we tested whether the progressively increasing levels of embryonic CHH methylation could be merely due to increased proportions of cotyledon tissues in embryos as they develop. Namely, we dissected cotyledon and non-cotyledon tissues from bent-cotyledon-staged embryos, and profiled their methylomes (Fig. 3h). Both cotyledon and non-cotyledon tissues were similarly hypermethylated on DMRs indicating that hypermethylation occurs throughout late-staged embryos and is not confined to the terminally differentiated cotyledons (Fig. 3i). Interestingly, the CHH hypermethylation observed in mature embryos resembled the hypermethylation reported in root columella cells and pollen vegetative nuclei [30, 50]. Similar to these specific cell-types, mature embryos have also exited the cell-cycle, which is further supported by increased and decreased levels of transcripts encoding negative and positive regulators of the cell-cycle, respectively (Additional file 2: Figure S3E) [51]. Therefore, CHH hypermethylation of mature embryos, as well as root columella cells and vegetative nuclei, is associated with cell cycle dormancy or exit.

### Small RNA-directed methylation of transposons during embryogenesis

To test whether 24-nt siRNAs derived from euchromatic and heterochromatic TEs were necessary for progressive TE methylation during embryogenesis, we performed methylC-seq on 24-nt siRNA-deficient early heart (4 DAP) and bent-cotyledon embryos (8 DAP), as well as leaves and floral buds (Additional file 1: Table S1). *NRPD1A* encodes the largest subunit of RNA polymerase IV (Pol IV) [16], and accordingly, 24-nt siRNAs overlapping euchromatic and heterochromatic TEs were nearly eliminated in *nrpd1a-3* mutants (Fig. 4a, c). Nearly all euchromatic TEs were completely hypomethylated in *nrpd1a* embryonic and post-embryonic tissues (Fig. 4b and Additional file 2:

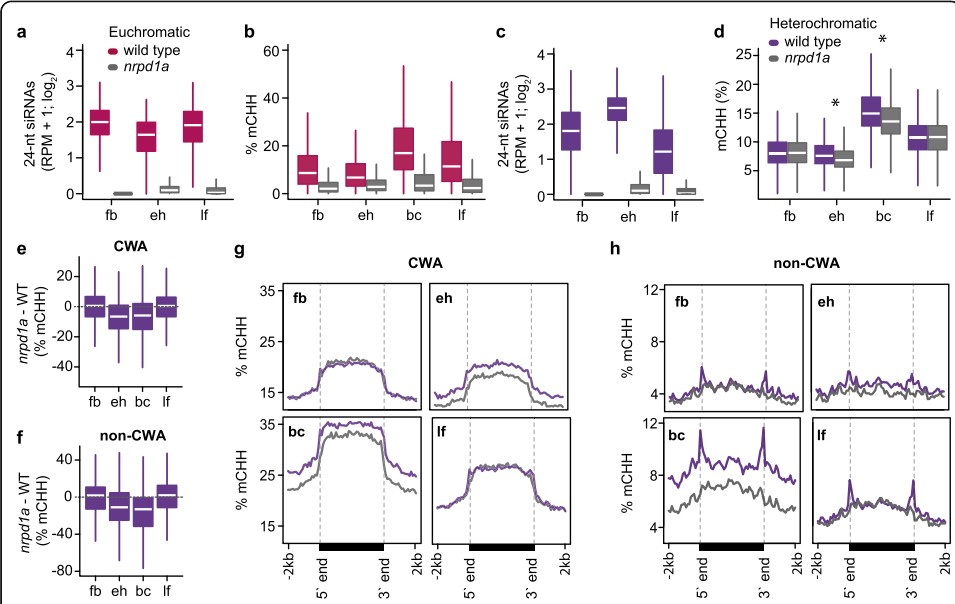

**Fig. 4** Small RNA-directed methylation of transposons during embryogenesis. **a, c** Boxplots of 24-nt siRNA levels in wild-type and *nrpd1a* mutant floral buds (fb), early heart embryos (eh), and leaves (lf) derived from euchromatic (**a**) or heterochromatic (**c**) TEs. Thick horizontal bars indicate medians, and the top and bottom edges of the box indicate the 75th and 25th percentiles, respectively. **b, d** Boxplots of CHH methylation levels in wild-type and *nrpd1a* mutant floral buds (fb), early heart embryos (eh), bent cotyledon embryos (bc), and leaves (lf) for euchromatic (**b**) or long heterochromatic (**d**) TEs. *P* values < 0.0001 based on Mann-Whitney *U* test of methylation differences between wild-type and *nrpd1a* tissues are represented by * in **d**. All differences displayed in **a**–**c** had *P* values < 0.0001. **e, f** Boxplots of CHH methylation differences between *nrpd1a* and wild-type tissues for long heterochromatic TEs in either CWA (**e**) or non-CWA (**f**) contexts. **g, h** Metaplots of average weighted CHH methylation percentages of long heterochromatic TEs in *nrpd1a* and wild-type tissues in either CWA (**g**) or non-CWA (**h**) contexts. See also Additional file 2: Figure S4

Figure S4). Because the 5729 heterochromatic TEs classified based on their embryonic siRNA dynamics also included short TEs methylated by the RdDM pathway in post-embryonic tissues (Additional file 2: Figure S2D, E), we partitioned heterochromatic TEs into either short (≤ 723 bp), medium (724–2114 bp), or long (> 2114) and examined their methylation levels in *nrpd1a* tissues (Fig. 4d and Additional file 2: Figure S5A-C). Consistent with previous observations from post-embryonic tissues [26, 27, 43], CHH methylation of short and medium TEs was significantly reduced compared to wild type in all *nrpd1a* tissues tested including embryos (Additional file 2: Figure S5B, C). In contrast, long heterochromatic TEs were globally unaffected in *nrpd1a* post-embryonic tissues, but significantly hypomethylated in *nrpd1a* mutant embryos relative to wild type (Fig. 4d). Methylation of long heterochromatic TEs is thus partially dependent on siRNAs in embryonic, but not post-embryonic tissues.

Long heterochromatic TEs are methylated by CMT2 in post-embryonic tissues, and their highly condensed chromatin states were proposed to inhibit siRNA-directed DRM2-mediated methylation [26, 27, 43, 52]. CHH methylation can be classified as CWA (W = A or T) or non-CWA. DRM2 methylates CWA and non-CWA sites, and CMT2 preferentially methylates CWA nucleotides [53–55]. Only the edges of long heterochromatic TEs were hypomethylated in non-CWA contexts of post-embryonic *nrpd1a* mutant tissues relative to wild type. However, both edges and bodies of long heterochromatic TEs were hypomethylated in all CHH contexts in early heart and

especially bent-cotyledon *nrpd1a* embryos compared to wild type (Fig. 4g, h). There-
fore, 24-nt siRNAs originating from both bodies and edges of long heterochromatic
TEs rapidly increase in early embryos and are partially required for TE methylation.

### Chromatin regulates small RNA transcription

Heterochromatin prevents access to de novo methyltransferases [27, 56], and thus, it
may also impede Pol IV access and resulting siRNA transcription. Consistent with rela-
tively low transcript levels of heterochromatin-promoting factors in early embryos [57],
reduction of DAPI-stained chromocenters and enlarged nuclei in zygotes compared to
somatic tissues indicated that zygotic chromatin is decondensed (Fig. 5a, b and
Additional file 2: Figure S6A). A marked increase in zygotic nucleoli size, as well as the
co-localization of hundreds of 5S rRNAs and heterochromatic TEs in centromeric re-
gions (Copenhaver, 1999; Simon et al., 2018), further suggested that heterochromatic
TEs could be decondensed in early embryos (Fig. 5b, c) Therefore, post-fertilization
heterochromatin decondensation, potentially associated with rRNA production, may
permit Pol IV accessibility to heterochromatic TEs and corresponding transcription of
24-nt siRNA precursors soon after fertilization. We then used a more experimentally
tractable post-embryonic tissue to test how chromatin may generally regulate siRNA
production.

To investigate the effects of H1 depletion on TE-derived siRNAs in more detail, we
Because linker histone 1 (H1) inhibits RNA polymerases from binding to chromatin
[60, 61], and its depletion results in the loss of both chromocenters and chromatin
compaction [59, 62, 63], we tested whether decreased H1 levels during post-
embryogenesis were sufficient to increase siRNA biogenesis throughout long hetero-
chromatic TEs similar to what we observed in embryos. That is, we performed sRNA-
seq on leaves carrying null mutations in the two expressed *H1.1* and *H1.2* isoforms
(i.e., *h1.1-1/h1.2-1* or *h1* mutants) and found that 24-nt siRNAs from heterochromatic
TEs were significantly increased by more than 3.7-fold and were predominantly derived
from TE bodies (Fig. 5d: Additional file 2: Figure S6B). Strikingly, hierarchical cluster-
ing of TE-derived 24-nt siRNAs demonstrated that *h1* leaf siRNA populations were
more similar to siRNA populations from wild-type mature embryos instead of leaves
indicating that H1 depletion was sufficient to induce an embryo-like siRNA population
in a post-embryonic tissue (Additional file 2: Figure S6C).

To investigate the effects of H1 depletion on TE-derived siRNAs in more detail, we
ranked euchromatic and heterochromatic TEs based on their 24-nt siRNA levels in *h1*
mutant leaves and found that increased 24-nt siRNA levels from long heterochromatic
TEs in *h1* mutants were positively correlated with TE length, CHH methylation, and
H1 occupancy (Fig. 5e–g and Additional file 2: Figure S6D). Because TEs with in-
creased 24-nt siRNAs in *h1* mutants were also enriched for HISTONE 3 LYSINE 9 di-
methylation (H3K9me2) (Fig. 5h), we tested whether loss of H3K9me2 can also affect
heterochromatic siRNA production. We examined the levels of siRNAs from long het-
erochromatic TEs in leaves triple mutant for *SU(VAR)3-9 HOMOLOG 4/5/6* histone
methyltransferases (*suvh4/5/6*), which are deficient in H3K9me2 levels (Additional file 1:
Table S1) [43]. Long heterochromatic TEs produced only 1.2-fold more 24-nt siRNAs
in *suvh4/5/6* leaves compared to wild type, and the global 24-nt siRNA populations
were similar to wild type (Additional file 2: Figure S6B, C). These results suggest that

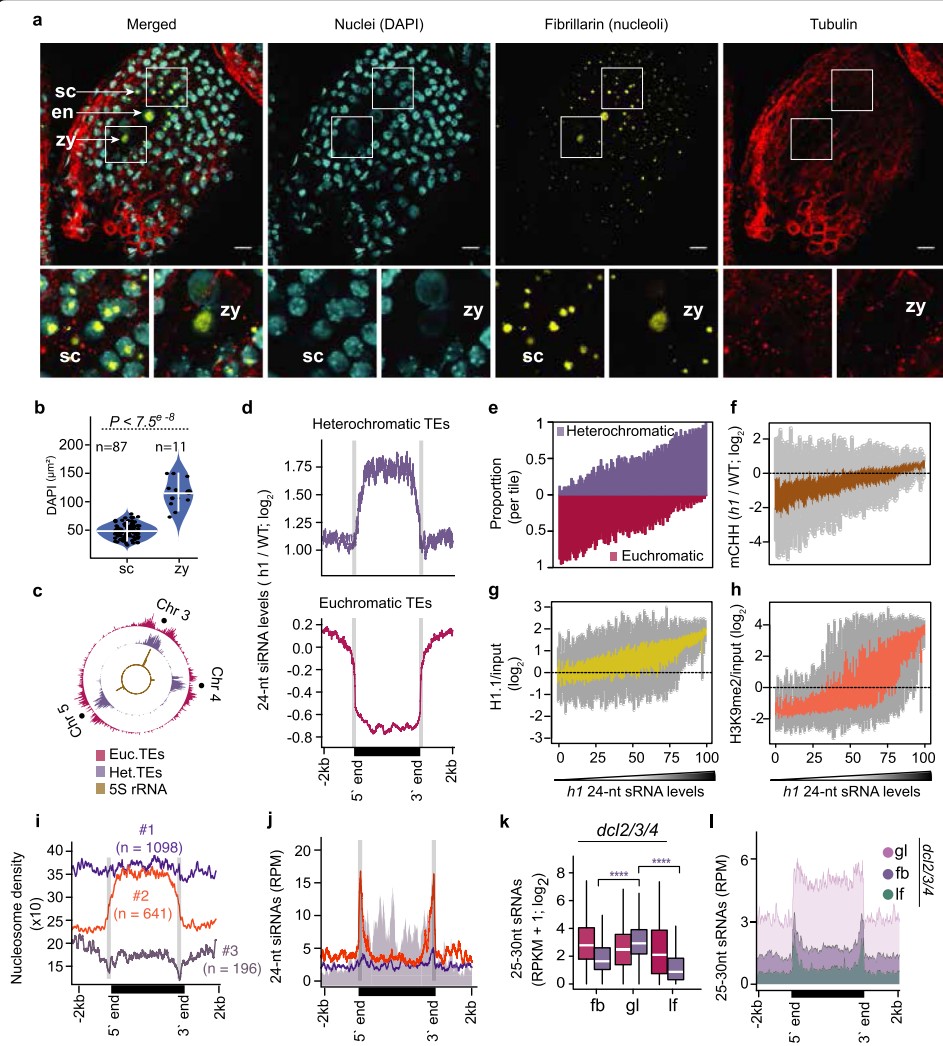

**Fig. 5** Chromatin regulates small RNA transcription. **a** Representative expansion microscopy images of DAPI-stained nuclei (cyan), fibrillarin (yellow), and tubulin (red) in zygote-containing seeds (*top*). Increased magnification of seed coat and zygote sections marked in white squares (*bottom*). zy, zygote; en, endosperm; sc, seed coat. Scale bars represent 20 μm. **b** Violin and dot plot representing the DAPI-stained area of seed coat (sc) and zygote (zy) nuclei. Horizontal and vertical white colored bars represent median and interquartile range, respectively. *P* values based on the Wilcoxon rank sum test of DAPI area differences between seed coat (sc) and zygote (zy) are shown. **c** Circos plot representing densities of euchromatic (red) and heterochromatic (purple) TEs, and 5S rRNA loci on chromosomes 3, 4, and 5. Centromeres are indicated by dots. **d** Metaplots of 24-nt siRNA levels in *h1* mutants relative to wild-type leaves for heterochromatic (*top*) or euchromatic (*bottom*) TEs. **e–h** TEs were divided into percentiles, ordered based on their 24-nt siRNA levels in *h1* mutants, and plotted according to proportion of euchromatic and heterochromatic TEs (**e**), CHH methylation levels in *h1* relative to wild type [27] (**f**), and relative enrichments of H1.1 (**g**) and H3K9me2 (**h**) [58]. **i, j** Metaplots displaying nucleosome occupancy of long heterochromatic TE groups 1–3 according to MNase-seq datasets [59] (**i**) or 24-nt siRNA levels (**j**). Annotated TE 5′ and 3′ ends are labeled and indicated by vertical gray lines. **k** Boxplots of 25–30-nt precursor siRNA levels of euchromatic (red) and heterochromatic (purple) TEs in *dcl2/3/4* floral buds (fb), globular embryos (gl), and leaves (lf). Thick horizontal bars indicate medians, and the top and bottom edges of the box indicate the 75th and 25th percentiles, respectively. *P* values < 0.0001 based on Mann-Whitney *U* test of differences between globular and floral bud or leaf 25-to-30-nt siRNAs derived from heterochromatic TEs (purple) tissues are represented by ****. **l** Metaplots of 25–30-nt siRNA levels overlapping long heterochromatic TEs in *dcl2/3/4* floral buds (fb), globular embryos (gl), and leaves (lf) (key). Annotated TE 5′ and 3′ ends are labeled at the bottom. See also Additional file 2: Figure S5

reducing heterochromatin, rather than H3K9me2 marks associated with heterochromatin, is sufficient for siRNA production from long heterochromatic TEs. Accordingly, H1 promotes nucleosome occupancy on heterochromatic regions [58, 59], and we found that heterochromatic TEs were enriched for H1 (Fig. 5g and Additional file 2: Figure S6D) and had reduced nucleosome occupancy in *h1* mutants (Additional file 2: Figure S6F). Our results indicate that depletion of H1 is sufficient to decrease nucleosome occupancy of heterochromatic TEs, as well as increase corresponding siRNA and CHH methylation levels.

Because 24-nt siRNAs were enriched on the bodies of long heterochromatic TEs in *h1* mutants with reduced chromatin compaction (Fig. 5d and Additional file 2: Figure S6B), we next examined the relationships between nucleosome occupancy and 24-nt siRNA levels of long heterochromatic TEs in wild-type leaves. We employed an iterative *k*-means clustering approach to generate three groups of long heterochromatic TEs based on their nucleosome occupancy using publicly available micrococcal nuclease sequencing data [59] (Fig. 5i). Group 1 comprised 1098 TEs (56.7% of total) that had high densities of nucleosomes and were devoid of 24-nt siRNAs throughout their lengths (Fig. 5i, j). Group 2 contained 641 TEs (33.1% of total) and had low and high nucleosome occupancy over the edges and bodies, respectively, and were enriched for 24-nt siRNAs only on the edges (Fig. 5i, j). Group 3 consisted of only 196 TEs (10.2% of total) and had very low nucleosome levels, but abundant 24-nt siRNAs, on both their edges and bodies similar to euchromatic TEs (Fig. 5i, j). Altogether, these results suggest that increased nucleosome occupancy restricts RNA Pol IV activity, and thus, chromatin states alone appear to explain 24-nt siRNA production from TEs.

RNA Pol IV transcribes ~ 25-to-40-nt RNAs that are co-transcriptionally converted to double-stranded RNAs by RNA-dependent RNA Polymerases and rapidly processed into 23-nt/24-nt duplexes by DICER-LIKE (DCL) endoribonucleases [17, 18, 22]. These transient Pol IV-dependent 24-nt siRNA precursors can be robustly detected in *dcl2/3/4* mutants [17, 18, 64], and thus, 24-nt siRNA precursor levels indicate Pol IV transcriptional activities. We performed sRNA-seq on *dcl2/3/4* flowers, globular embryos, and leaves and compared levels of 24-nt siRNA precursors from euchromatic and heterochromatic TEs (Additional file 1: Table S1). Compared to leaves and flowers, we respectively detected 11-fold and 3.8-fold significantly more 24-nt precursors from heterochromatic TEs in *dcl2/3/4* early embryos (Fig. 5k). Importantly, the 24-nt siRNA precursors mostly originated from the bodies of long heterochromatic TEs in embryos, but were strongly reduced in floral buds and leaves (Fig. 5l). Together with the observations that 24-nt siRNAs were also enriched on the bodies of long heterochromatic TEs in early embryos and *h1* mutant leaves (Figs. 2c, and 5d), our results are consistent with a model whereby decompaction of heterochromatin in early embryos and *h1* mutant leaves permits Pol IV access and transcriptional activities to produce 24-nt siRNAs.

### Homeostasis of transposon-derived siRNAs

In contrast to heterochromatic TEs, we found 2.7-fold significantly less 24-nt siRNAs from euchromatic TEs in *h1* leaves compared to wild type, which was also associated with their CHH hypomethylation (Fig. 5d, f and Additional file 2: Figure S6B). Unlike heterochromatic TEs, euchromatic TEs were lowly enriched for H1 in wild-type leaves

and nucleosome occupancy was further reduced in *h1* mutants (Fig. 5g and Additional file 2: Figure S6D, F). Therefore, siRNA depletion from euchromatic TEs is likely an indirect consequence of sequestering Pol IV to accessible heterochromatic TEs. Moreover, we observed the greatest enrichment of siRNAs derived from heterochromatic compared to euchromatic TEs at the preglobular stage of embryogenesis, which is the earliest post-fertilization sRNA-seq dataset available (Fig. 6a). This was reduced during mid-embryogenesis and then further decreased to almost post-embryonic levels during maturation (Fig. 6a). Together with our siRNA precursor analysis (Fig. 5k, l), this indicates that Pol IV is more efficiently recruited to heterochromatic TEs compared to euchromatic TEs during the initial stages of embryogenesis. The enrichment of heterochromatic TE-derived 24-nt siRNAs in preglobular embryos surpassed what we observed in *h1* mutant leaves (Fig. 6a), suggesting that reduced nucleosome occupancy alone does not fully account for the extreme enrichment of heterochromatic siRNAs in preglobular embryos.

Based on available sRNA-seq datasets, euchromatic, but not heterochromatic, TE-derived 24-nt siRNAs were substantially reduced in mutants deficient in CG and CHH methylation (Fig. 6b and Additional file 2: Figure S7) [21, 43, 69]. Therefore, low CHH

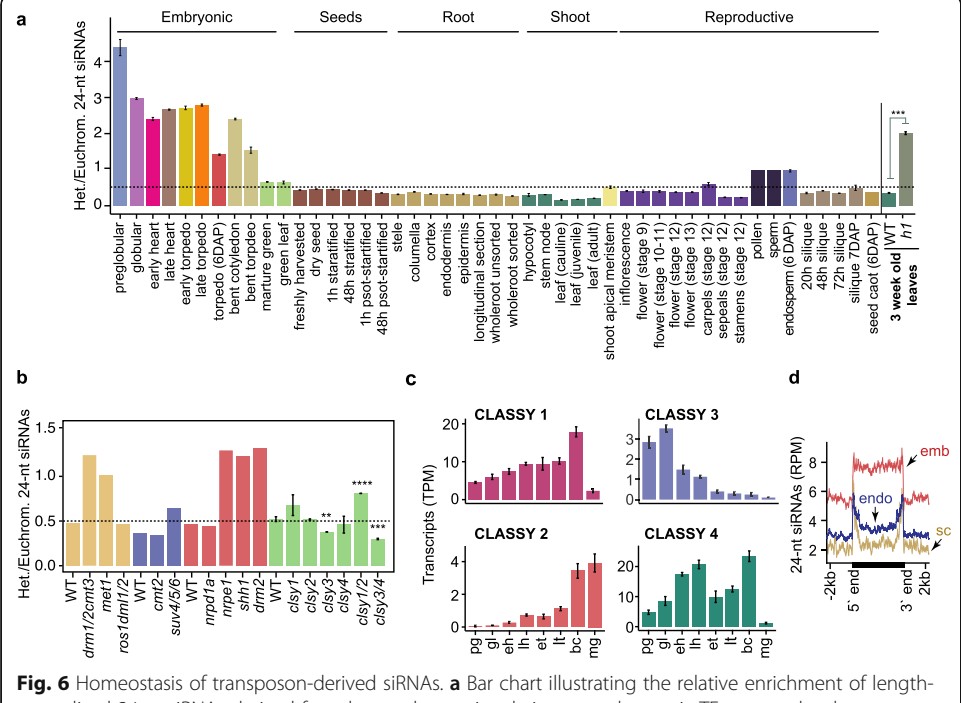

**Fig. 6** Homeostasis of transposon-derived siRNAs. **a** Bar chart illustrating the relative enrichment of length-normalized 24-nt siRNAs derived from heterochromatic relative to euchromatic TEs across development, and in *h1* mutant leaves. *P* values < 0.001 based on Student's *t* test of heterochromatic vs euchromatic 24-nt siRNA enrichment differences between wild-type and *h1* tissues are represented by ***. Various tissues and cell-types from different phases of the plant life-cycle are labeled and include those from published datasets [29, 39, 65–68]. **b** Relative enrichments of 24-nt siRNAs derived from heterochromatic relative to euchromatic TEs in RdDM and related mutants. Wild type (WT), *drm1/2 cmt3*, *met1*, and *ros1 dml1/2* [69]; WT, *cmt2*, and *suv4/5/6* [43]; WT, *nrpd1a*, *nrpe1*, *shh1*, and *drm2* [21]; WT, *clsy1*, *clsy2*, *clsy3*, *clsy4*, *clsy1/2*, and *clsy3/4* [19]. *P* values < 0.0001, < 0.001, and < 0.01 based on Student's *t* test of heterochromatic vs euchromatic 24-nt siRNA enrichment differences between wild-type and *mutants* are represented by ****, ***, and **. **c** CLASSY1, CLASSY2, CLASSY3, and CLASSY4 transcript levels (TPM) during embryogenesis. **d** Metaplots of 24-nt siRNA levels overlapping long heterochromatic TEs in 6 DAP embryo (red; emb), endosperm (blue; endo), or seed coat (yellow; sc) tissues based on [65]. See also Additional file 2: Figure S6

methylation in preglobular embryos (Fig. 3a, b) may reduce methylation-dependent feed-back loops that facilitate production of siRNAs from euchromatic TEs in preglobular stages. CLASSY (CLSY) chromatin remodeling factors also promote 24-nt siRNA production: CLSY 1/2 and CLSY 3/4 help recruit Pol IV to euchromatic and heterochromatic regions, respectively (Fig. 6b) [19, 21, 43, 69]. Dynamic chromatin states and corresponding establishment of methylation/CLSY-dependent transcription of siRNA precursors likely contribute to the unique siRNA populations observed in embryos.

Remarkably, the relative amounts of euchromatic and heterochromatic TE-derived siRNAs remained stable throughout post-embryonic development (Fig. 6a). Root columella and pollen vegetative cells are depleted for H1 [50, 70], but were not depleted for euchromatic siRNAs (Fig. 6a). This may be due to these terminally differentiated cell-types being derived from a single mitotic division, and thus, they may retain the ability to recruit Pol IV to euchromatic TEs. For example, these cell types are CHH hypermethylated [30, 50], and thus, CHH methylation-dependent positive feedback loops on euchromatic TEs may counteract the loss of H1. Importantly, TE-derived siRNA populations in embryos were distinct from endosperm and seed-coat tissues (Fig. 6d). Consistent with chromatin states primarily regulating siRNA production, the endosperm has reduced cell division compared to embryos by 6 DAP [71]. Accordingly, endosperm siRNAs overlapped mostly edges, but not bodies, of heterochromatic TEs typical of other non-embryonic populations (Fig. 6d). Similar to pollen, but in contrast to other non-embryonic tissues, endosperm had similar levels of siRNAs from heterochromatic and euchromatic TEs (Fig. 6a). Because nearly equal siRNA levels from euchromatic and heterochromatic siRNAs were also observed in *met1* mutants (Fig. 6b and Additional file 2: Figure S7), this balance may be due to loss of CG methylation-dependent euchromatic siRNA production in CG hypomethylated pollen vegetative nuclei and endosperm [32]. Altogether our data indicate that the homeostasis of 24-nt siRNA production from euchromatic and heterochromatic TEs are affected by dynamic chromatin states including, but not restricted to, those associated with early embryogenesis.

## Discussion

Although siRNAs direct faithful re-establishment of methylation genome-wide across generations [72], the dynamics of embryonic siRNAs and how they contribute to the nascent epigenome have not been reported. In this study, we demonstrated that thousands of TEs produce exceptionally high levels of 24-nt siRNAs in embryos (Fig. 1) and can be classified into two distinct groups based on their developmental dynamics (Fig. 2). siRNAs from euchromatic TEs gradually increase to post-embryonic levels during embryogenesis and are constitutively required to direct TE methylation in embryonic and post-embryonic tissues (Fig. 4). In contrast, heterochromatic TEs produce a burst of siRNAs soon after fertilization, and specifically during embryogenesis, to help establish TE methylation de novo, which is then maintained independent of siRNAs during post-embryogenesis (Fig. 4) [26, 27, 43]. Interestingly, the levels of siRNAs from these euchromatic and heterochromatic bipartite-classified TEs are regulated according to their chromatin states (Fig. 5). Decondensed chromatin permits transcription of 24-nt siRNAs, and this contributes to cell autonomous homeostatic control mechanisms that normalize chromatin states.

We propose a three-phase model for how chromatin states, and resulting siRNA dynamics, help shape the nascent epigenome (Fig. 7). After fertilization, zygotic chromatin is decondensed and this appears to be associated with transcriptional activation of rRNAs, including hundreds of 5S rRNA loci that co-localize with heterochromatic TEs near centromeric regions (Fig. 5a, c). Arabidopsis zygotes require de novo synthesis of gene products directly after fertilization [57, 73], and genes involved in rRNA biogenesis produce high levels of transcripts in preglobular embryos relative to later stages [40]. In contrast to Arabidopsis, maternally donated proteins drive early embryogenesis in Xenopus and H1 dynamics mediate transcriptional activation of rRNA loci in oocytes and their silencing in somatic tissues [74, 75]. In pollen vegetative cells, decondensation of rRNA loci can also permit their transcription and concomitant cell growth [76] and may be required for the rapid cell divisions in early endosperm, which also have enlarged nucleoli (Fig. 5a). Therefore, reduced heterochromatin in a variety of cell types, including those producing large amounts of protein such as early embryos, endosperm, and pollen vegetative cells, may be permissive for Pol IV-mediated transcription of siRNA precursors from TEs that are typically in a deep heterochromatic state during other developmental phases (Fig. 7; phase 1). Consistent with decondensation of heterochromatin facilitating de novo production of sRNAs, Pol IV-dependent 24-nt siRNAs were sharply increased throughout heterochromatic TEs in *h1* mutant leaves with reduced heterochromatin (Fig. 5d and Additional file 2: Figure S6). Production of siRNAs from euchromatic TEs is delayed relative to those from heterochromatic TEs (Fig. 6a). Because euchromatic, but not heterochromatic, TEs require CHH methylation and CLSY1/2 chromatin remodelers to produce full siRNA levels (Fig. 6b) [19, 21, 43, 69], both of which are gradually increased during embryogenesis (Figs. 3a–c and 6c), the developmental time-lag in euchromatic compared to heterochromatic siRNA production may also be partially due to the delay establishing methylation/CLSY-dependent positive feedback loops during early embryogenesis.

Embryos divide rapidly through the bent cotyledon stage [77] and dynamic chromatin condensation and decondensation associated with such increased cell division [78, 79] likely allows access of Pol IV to both heterochromatic and euchromatic TEs (Fig. 7;

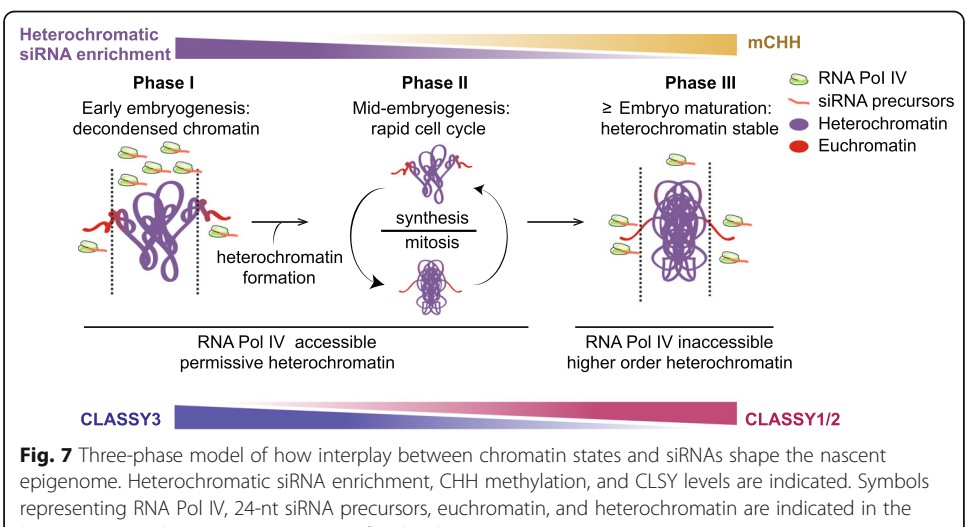

**Fig. 7** Three-phase model of how interplay between chromatin states and siRNAs shape the nascent epigenome. Heterochromatic siRNA enrichment, CHH methylation, and CLSY levels are indicated. Symbols representing RNA Pol IV, 24-nt siRNA precursors, euchromatin, and heterochromatin are indicated in the key. See text in the "Discussion" section for details

Phase 2). Production of embryonic siRNAs from heterochromatic relative to euchromatic TEs is steady between the globular and bent cotyledon stages. Heterochromatic TE-derived siRNAs are rapidly reduced upon maturation when heterochromatin becomes highly condensed [80] and euchromatic domains containing genes encoding seed storage and oil body biogenesis proteins are transcriptionally activated [40] (Fig. 6a). As a consequence, Pol IV access to heterochromatic TEs is likely greatly reduced, and this results in more Pol IV being readily available to produce siRNAs from euchromatic TEs (Fig. 7; Phase 3). Consistently, we observed a burst of siRNAs from euchromatic TEs, and their associated hyper methylation, at the mature stage (Fig. 2a, b). Heterochromatic TEs also become hypermethylated at the mature stage, which appears to be largely independent of siRNAs, but rather dependent on CMT2 as is typical of subsequent post-embryonic development (Fig. 4). Therefore, CHH hypermethylation throughout mature embryos may largely be a consequence of the rapid shift of chromatin states upon maturation.

Similar to *h1* mutants [58, 59], heat stress also causes reduced nucleosome occupancy and decondensed heterochromatin [81, 82]. Moreover, we classified the heat-activated ONSEN/AT5TE15240 as a class B/heterochromatic TE based on its siRNA dynamics, and ONSEN transcription and transposition are greatly enhanced in RdDM-defective mutants [83]. Interestingly, 24-nt siRNAs were increased throughout the body of ONSEN TEs after heat stress, and this was further enhanced in rapidly dividing undifferentiated calli [84], which has increased chromatin accessibility in rice [84, 85]. Based on these results, as well as our previous observation that heat-stress-related genes are significantly enriched in preglobular embryos [40], we suggest that the chromatin dynamics caused by heat stress and subsequent recovery are analogous to what we observed in early embryos. That is, heat stress and fertilization may decrease nucleosome occupancy across heterochromatic TEs, which enables Pol IV-mediated siRNA production and subsequent reconstitution of proper heterochromatin to help limit TE mobilization in the genome. Strong upregulation of TE-specific endo-siRNAs and piwi-interacting RNAs observed in H1-depleted Drosophila [86] suggests that H1-dependent regulation of chromatin states may also facilitate heterochromatic small RNA transcription in animals.

## Conclusions

Reprogramming of heterochromatin during early embryogenesis occurs in diverse metazoa including flies, mammals, worms, and zebrafish [8–12], and we observed that CHH methylation is also reprogrammed during early plant embryogenesis (Fig. 3). For example, CHH methylation was essentially lost on the bodies of heterochromatic TEs in sperm and subsequently fully re-established by both siRNA-dependent and siRNA-independent pathways (Fig. 4). Reduced heterochromatin in early animal embryos has been associated with increased developmental potential [87, 88], and similar relationships have also been observed in plants including reprogramming associated with plant regeneration and heat-stress induced somatic embryogenesis [89–95]. However, decreased heterochromatin would also increase the risk of TE mobilization and resulting mutations, and this could be especially dangerous in plant zygotes because they are the precursors of all cell types including the gametes. Our results indicate that embryos produce 24-nt siRNAs according to their chromatin states including those that are

permissible for Pol IV transcription soon after fertilization. These de novo produced 24-nt siRNAs direct re-methylation of both euchromatic and heterochromatic TEs in the new generation. Therefore, decondensed chromatin permits transcription of early embryonic siRNAs to help promote cell-autonomous TE silencing. More generally, reduced heterochromatin due to sharp increases in rRNA production requirements during growth (e.g., early embryos, endosperm, and pollen vegetative cells), and perhaps in response to external cues such as heat stress, enables the synthesis and functions of sRNAs that can help reconstitute proper chromatin states.

## Methods
### Plant material and growth conditions
All genotypes were in the Columbia-0 (Col-0) *Arabidopsis thaliana* background including *dcl2/3/4* mutants composed of *dcl2-1*, *dcl3-1* and *dcl4-2t* [96], *h1.1-1/h1.2-2* [27], *nrpd1a-3* [97], and *suv4/5/6* [98]. Plants were grown in a climate-controlled growth chamber at 20 to 22 °C under a 16-h light/8-h dark cycle with incandescent lights at 130 to 150 μmol/m$^2$/s.

### Embryo isolation and nucleic acid extraction
Embryos were dissected from siliques either 3 days after pollination (DAP) (preglobular), 4 DAP (early heart/transition), or 8 DAP (bent cotyledon). Siliques were opened with forceps and seeds were collected in 2-ml Eppendorf tubes containing nuclease-free water and kept on ice. Seeds were then crushed with pestles, and embryos were selected under an inverted microscope using a microcapillary tube. Isolated embryos, as well as cotyledon and non-cotyledon portions of bent-cotyledon embryos, were thoroughly and serially washed 4× with nuclease-free water and stored at − 80 °C. RNA was isolated as previously described [39, 99]. Genomic DNA was extracted from ≥ 50 embryos per stage, floral buds, and leaves using *Quick*-DNA™ Micro prep Kit (Zymo D3020) according to the manufacturer's recommendations.

### Small RNA profiling
sRNA-seq libraries were generated as previously described [39]. Briefly, total RNA from each sample was size selected for 18 to 30-nucleotide RNAs using denaturing polyacrylamide-urea gels. Size-selected RNA was used to ligate adapters and synthesize cDNA with the NEBNext Multiplex Small RNA Library Prep Set for Illumina kit (cat. no E7300; New England Biolabs) according to the manufacturer's recommendations. Various numbers of PCR cycles were used to amplify cDNAs: 18, 20, 22, and 24 PCR cycles for globular and 14, 16, 18, and 20 PCR cycles for early heart and 3-week-old leaf samples. Final PCR amplicons were initially run on a 90% (v/v) formamide/8% (w/v) acrylamide gel for 30 min at 5 W, followed by 30 W for ≥ 2 h, and stained with SYBR Gold (1:10,000; Thermo Fisher Scientific). PCR amplicons between 137 and 149 bp corresponding to 18- to 30-nucleotide sRNAs with adapters, respectively, were inspected under a UV transilluminator, and amplicons with non-saturated signals generated from PCR cycles were gel-purified. Gel-purified small RNA libraries were resuspended in 15 μL of Elution Buffer (Qiagen). Finally, small RNA libraries were quality checked for the expected size range with Agilent High sensitivity NGS fragment Kit (DNF-474-

1000) and were sequenced on a HiSeq 2500 instrument (Illumina) in 50-base single-end mode.

### Small RNA sequencing analysis

Small RNA-seq library datasets generated in this study or downloaded from NCBI's Sequence Read Archive (SRA) were subjected to the same small RNA analysis pipeline. First, raw fastq files were adapter trimmed with *Cutadapt* [100] and sequences between 18 and 30 bases in length, and that contained an adapter were retained. The trimmed sequences were then aligned to the *Arabidopsis thaliana* TAIR10 genome [101] with STAR [102] requiring zero mismatches and allowing up to 100 multiple end-to-end alignments. Multi-mapping reads from aligned SAM files were re-assigned with a "rich-get-richer" algorithm using the custom python script "readmapIO.py" as described previously [103]. Resulting output bedFiles were then sorted, condensed, and normalized for total genome-matching reads. The BEDtools [104] map function was then used to quantify the sum of the normalized reads per million (RPM) mapping to TAIR10 annotated Transposable elements (TEs).

For the model-based clustering of transposon-derived 24-nt siRNAs, mean RPM of 24-nt siRNAs from biological triplicates of floral bud, embryonic, and leaf samples mapping to TEs were calculated and used as input for *R* library *Mclust* [42] to identify the optimal Gaussian mixture model (GMM). By employing *Mclust* function *mclustBIC(.,G=seq(2,20),by=2)* in sequential increments of two until twenty components, we identified the VEV (Variable volume, Equal shape, Variable orientation) ellipsoidal distribution model to be optimal with the minimum number of components (i.e., eight) containing maximum Bayesian Information Criterion (BIC). Finally to yield eight transposon clusters with VEV ellipsoidal distribution model, the *Mclust(.,G=8, modelNames="VEV")* function was applied on mean TE-derived 24-nt siRNAs.

Principal component analysis of 24-nt siRNAs was performed with the R *prcomp* function using default parameters. Hierarchical clustering of transposon-derived 24-nt siRNAs was performed by calculating Euclidean distances between samples and the distance matrix was subjected to the R function *hclust(\*,"complete")*. Heatmaps and metaplots of TE-derived siRNAs were generated with *deepTools* [105]. Briefly, a matrix containing normalized 24-nt siRNA scores per genome regions for tissue types or genotypes were generated (*computeMatrix scale-regions -bs 5 -m 4000 -b 2000 -a 2000 --averageTypeBins mean*). The obtained matrix was used to generate heatmaps (*deepTools plotHeatmap*) or metaplots (*deepTools plotProfile*). For Fig. 5, regions without siRNA signals were removed and the remaining genomic regions were used to calculate matrix containing nucleosome signal and 24-nt siRNA levels. This matrix then served as input to employ Iterative K-means clustering with *deepTools* function *plotProfile --kmeans*.

### DNA methylation profiling

MethylC-seq libraries were generated using post-bisulfite adapter tagging (PBAT) to avoid the bisulfite-induced loss of intact sequencing templates as described [48] with the following modifications. Briefly, genomic DNA was subjected to bisulfite treatment for 200 min with EZ DNA Methylation-DirectTM Kit (Zymo D5020). Bisulfite-treated

DNA was then preamplified for two cycles with primers (5′-CCCTACACGACGCTCT TCCGATCTNNNNNNN-3′) containing random hexamers and purified using the Zymo DNA Clean and Concentrator kit. Adaptor primers (5′-CAGACGTGTGCTCTTCCG ATCTNNNNNNN-3′) were added to preamplified products and then amplified for 12 PCR cycles with indexing primers for Illumina sequencing. Methylome libraries were purified using Beckman Coulter AMPureXP DNA beads. Libraries quality checked for fragment length between 200 and 600 bp were used for sequenced in single-read mode on an Illumina HiSeq2500 or Nextseq instrument.

### DNA methylation analysis

Sequenced reads were quality filtered and trimmed using *Trim Galore* with default settings. In addition, the first six bases of each read were removed to exclude random hexamers from the pre-amplification step of library construction and to also reduce 5′ methylation-bias (m-bias). Reads were aligned against the C-to-T converted TAIR10 genome using *Bismark* in non-directional mode to original top strand (OT), original bottom strand (OB), complementary to OT (CTOT) and OB (CTOB) (*bismark --non_directional -q --score-min L,0,-0.4*) [106]. Aligned BAM files containing clonal duplicates were removed with function *deduplicate_bismark -s --bam*, and uniquely mapped reads were then used as input for the *Methylpy* software [107]. Weighted methylation rates at each covered cytosine was extracted using command *methylpy call-methylation-state --paired-end FALSE.* Bisulfite conversion rates were calculated using the unmethylated chloroplast genome or spiked-in unmethylated Lambda phage DNA controls (European Nucleotide Archive Accession Number J02459, Promega catalog number D1521). FASTQ files obtained from publicly available methylomes generated from sperm [32], early torpedo [49], mid-torpedo to early maturation [31], mature green [37] embryos, and H1 mutant tissues [27] were also processed in the similar manner; except only 5′ end nucleotides of the reads with m-bias were removed and aligned in directional mode to OT and OB strands.

Differentially methylated regions (DMRs) were defined using *Methylpy* as described [36]. Briefly, biological replicates were pooled and differentially methylated sites (DMSs) were identified by the root mean square tests with false discovery rates ≤ 0.01. Cytosine sites with ≥ 4 overlapping reads were retained for all samples except for preglobular in which DMSs with ≥ 3 overlapping reads were retained. Differentially methylated sites within 100-bp were collapsed into DMRs. CHH-DMRs were further filtered by discarding regions with < 4 DMSs and methylation differences < 20%. Using these parameters, DMRs were identified in all 10 pairwise combinations across embryonic samples (preglobular, early heart, early torpedo, bent cotyledon, mature green) and merged using the BEDtools *merge* function [104]. DMRs were used to calculate the weighted CHH methylation rate on all analyzed tissue types. CHH methylation meta-plots for class A, B, and siRNA-deficient TEs were plotted using the R library *Seqplots* [108]: Body, upstream, and downstream regions of TEs were split into equal-sized bins, and the average weighted mCHH level for each bin was calculated and plotted.

### Expansion microscopy and DAPI quantification

The expansion microscopy technique [109] optimized for Arabidopsis seeds was conducted as previously described [73]. Anti-Fibrillarin antibody (ab4566, Abcam) and

anti-alpha Tubulin antibody (ab89984, Abcam) were used in 1:500 dilution as primary antibodies. Goat Anti-Mouse IgG H&L (Alexa Fluor® 488) (ab150113, Abcam) and goat Anti-Chicken IgY H&L (Alexa Fluor® 555) (ab150170, Abcam) were used in 1:500 dilution as secondary antibodies. For each sample, a stack of nine images with 1-μm intervals were recorded by ZEISS LSM700 with 25× oil objective and ZEN software at $1024 \times 1024$ resolution in 8-bit. DAPI signals were excited by 405-nm laser and passed through SP490 filters. Alexa488 signals were excited by 488-nm laser and passed through BP490-635 filters. Alexa555 signals were excited by 555-nm laser and passed through 560–1000-nm filters. Pinhole sizes were kept as 1 airy unit for each color, and color channels were scanned separately. FIJI software was used for image processing and nuclear size quantification. Each stack of images was first Z-projected on maximum intensity and then the nuclear areas were determined based on DAPI signals. The zygotic nuclei were distinguished from the endosperm nuclei according to position and tubulin staining patterns.

## Supplementary information

**Additional file 1: Table S1.** Datasets and general mapping statistics.

**Additional file 2: Figure S1.** Characteristics of embryonic 24-nt siRNAs and their similarities across samples. **Figure S2.** siRNA dynamics and characteristics. **Figure S3.** Benchmarking low-input methylC-seq, methylomes and cell-cycle transcripts. **Figure S4.** Genome browser screenshots of 24-nt siRNAs and DNA methylation levels. **Figure S5.** Size-based partitioning of heterochromatic TEs and small RNA-directed methylation. **Figure S6.** TE-derived siRNA accumulation and association with chromatin. **Figure S7.** TE-derived siRNAs in methylation mutants.

**Additional file 3: Table S3.** Euchromatic and heterochromatic TE classifications.

**Additional file 4: Table S4.** Differentially methylated regions.

**Additional file 5.** Review history.

#### Acknowledgements
We thank the Vienna Biocenter Core Facilities GmbH (VBCF) Next Generation Sequencing and Plant Sciences Facilities for next-generation sequencing and plant growth chamber access, respectively, and the Institute of Molecular Pathology-Institute of Molecular Biology-Gregor Mendel Institute Molecular Biology Services for instrument access and support. We also thank Alexander Vogt for help in optimizing low-input methylC-seq library preparation; Anna Smolka for technical assistance; Patrick Hüther and Claude Becker for advice on methylation analysis; Zdravko Lorkovi, Michael Borg, and Frédéric Berger for sharing reagents; and Michael Schon, Balaji Enugutti, and other members of the Nodine lab for valuable input.

#### Peer review information

#### Review history
The review history is available as Additional file 5.

#### Authors' contributions
R.K.P. and M.D.N. conceived the project; R.K.P. developed the methodology, implemented software used, and performed formal analysis; R.K.P., S.P., K.P., P.K., S.L., and M.D.N. conducted the experiments; R.K.P. and M.D.N. wrote and edited the article; M.D.N. supervised the project and acquired funding. The authors read and approved the final manuscript.

#### Funding
This work was supported by the European Research Council under the European Union's Horizon 2020 Research and Innovation Program (grant 637888 to M.D.N.).

#### Availability of data and materials
All sequencing data generated in this study are available at the National Center for Biotechnology Information Gene Expression Omnibus (NCBI GEO, https://www.ncbi.nlm.nih.gov/geo/) under accession number GSE152971 [110]. Publicly available next-generation sequencing data were downloaded from NCBI, GEO, and are listed along with general mapping statistics in Additional file 1: Table S1. The software code used for the sRNA, methylome, and transcriptome analysis including a nextflow pipeline is available at https://github.com/mnodine/Papareddy.2020 [111]. BigWig

files of processed datasets generated either as part of this study or publicly available can be downloaded at https://github.com/mnodine/Papareddy.2020/tree/master/processed_BigWigs [111].

**Ethics approval and consent to participate**
Not applicable.

**Consent for publication**
Not applicable.

**Competing interests**
The authors declare that they have no conflicts of interests.

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

## 

