## [**Additional file 5.** Review history. · Genome Biology]

Review History

First round of review

Reviewer 1

Are you able to assess all statistics in the manuscript, including the appropriateness of statistical tests used? Yes, and I have assessed the statistics in my report.

Comments to author:

The manuscript by Papareddy et al. uses the model plant *Arabidopsis thaliana* to investigate the regulatory role of chromatin in the production of small RNAs (sRNA) during plant embryogenesis. Current models suggest that the hypomethylation-mediated reactivation of TE transcription in companion cells serve as substrates for the production of sRNAs which then move into gametes and embryo to silence TEs. Based on their findings, the authors suggest a cell-autonomous mechanism by which TE-derived sRNA facilitate the re-methylation of TEs throughout embryogenesis. By using a low-input small RNA sequencing (sRNA-seq) method that they developed before, they profiled the sRNA population during eight stages of *Arabidopsis* embryogenesis and focused on highly abundant TE-derived 24-nt long siRNAs. Their principal component analysis in Fig.1c nicely illustrates the developmental differences among the different stages and also the differences between leaf, mature and premature embryos. An inspection of the TEs that correspond to the developmentally dynamic 24-nt siRNAs identified two main classes of TEs (Class A and B) showing clear differences with regard to TE length and more importantly chromatin state. The authors also generated methylomes from preglobular, early heart, bent-cotyledon embryos, leaves, and floral buds. They discovered a clear association of 24-nt siRNAs with differentially methylated euchromatic and heterochromatic TEs during embryogenesis. In addition, by profiling the linker histone 1 h1 mutant, the authors found that the chromatin state alone can explain 24-nt siRNA production from TEs. The presented manuscript is clearly written and well-organized. The experiments/data analyses are logical and well-described and the findings should be of interest to a broad audience. Overall the manuscript is in good shape and should be considered for publication in *Genome Biology*. I have only a couple of suggestions for improvement.

1. It would be nice to see some of the developmental DMRs in a genome browser screenshot to get a better idea of the differences between the profiled developmental stages. It also helps to assess data quality. Ideally, the authors should provide a genome browser link that includes the newly generated datasets as well as the used publicly available datasets for the reviewers so that the quality of their data can be directly inspected.
2. The authors use the term chromatin state in a traditional way (euchromatin and heterochromatin) which is correct but is in my opinion a bit confusing since many readers including myself associate chromatin states with a more detailed classification based on various chromatin features (histone mods, etc.). The authors should clarify that.
3. The title needs to be improved. It sounds too general and should better reflect the findings
4. Little issue with some references on page 14,
" Based on available sRNA-seq datasets, euchromatic, but not heterochromatic, TE-derived 24-nt siRNAs

were substantially reduced in mutants deficient in CG and CHH methylation (Fig. 6b and Fig. S6a) (Law et al., 2013; Lister et al., 2008; Stroud et al., 2014)."

Reviewer 2

Are you able to assess all statistics in the manuscript, including the appropriateness of statistical tests used? Yes, and I have assessed the statistics in my report.

Comments to author:

The manuscript titled "Chromatin Regulates Bipartite-Classified Small RNA Expression to Maintain Epigenome Homeostasis in Arabidopsis" analyzed the dynamics of siRNAs during embryo development in Arabidopsis. The authors found that siRNA expression can be clustered into two main populations, one from TEs in euchromatic regions and the other from TEs in heterochromatic (peri-centromeric) regions. By comparing the difference in siRNA levels from the two types of TEs and correlate that with DNA methylation levels, the authors argue that the compactness of chromatin is a main determinant of the siRNA levels during embryo development. They tested this hypothesis in vegetative tissue using mutants defective in chromatin compaction or specific epigenetic modifications and found that loss of the linker histone H1 results in elevated siRNA levels from TEs in peri-centromeric regions. They thus proposed that chromatin decondensation facilitates Pol IV activity during early embryogenesis, which functions to promote heterochromatin homeostasis and TE silencing. Overall, I think the main conclusions are supported by the data. The writing and data presentation are clear. Importantly, this work is one of the few that study the epigenomic changes during embryo development; the view of cell autonomous regulation of heterochromatin during embryo development should be of interest to a wide range of plant biologists.

With that said, I do have a few minor suggestions:

1. Methods section should include more details considering that data analysis a main part of this paper. The current description of sRNA and DNA methylome analyses are too simple; how data from this study are compared to the published data should be included.
2. Page 10, 1st paragraph, Line 4-6. I don't think figure 4d represents long heterochromatic TEs.
3. Page 10, 2nd paragraph, last 3 lines. I don't quite understand what the sentence means.
4. Page 12, last paragraph, line 3. The reference by Yang DL et al. 2016 should also be included for the characterization of siRNA precursors.
5. Abbreviation of RNA polymerase IV should be "Pol IV" instead of "PolIV".
6. GEO accession number should be provided to the reviewers.

Response to Reviewer Comments

Thank you for taking the time to review our manuscript especially during the current COVID-19 pandemic. We are pleased with the broadly favorable reviewer comments, and have addressed each of them in the revised manuscript as detailed in the sections below. We have also included a version of the manuscript with changes highlighted and added live NCBI GEO accession numbers to the Availability of Data and Materials section of the revised manuscript, as well as editable figures and properly formatted text. Please note that we have now included Stefan Lutzmayer as an author on our manuscript because he generated sRNA-seq libraries that have not been published yet. We appreciate your constructive comments, and believe they have improved our manuscript. Please find the reviewer comments in black on the following pages, as well as our responses indicated in blue. We hope that you will now find it suitable for publication in *Genome Biology*.

Reviewer #1

The manuscript by Papareddy et al. uses the model plant Arabidopsis thaliana to investigate the regulatory role of chromatin in the production of small RNAs (sRNA) during plant embryogenesis. Current models suggest that the hypomethylation-mediated reactivation of TE transcription in companion cells serve as substrates for the production of sRNAs which then move into gametes and embryo to silence TEs. Based on their findings, the authors suggest a cell-autonomous mechanism by which TE-derived sRNA facilitate the re-methylation of TEs throughout embryogenesis. By using a low-input small RNA sequencing (sRNA-seq) method that they developed before, they profiled the sRNA population during eight stages of Arabidopsis embryogenesis and focused on highly abundant TE-derived 24-nt long siRNAs. Their principal component analysis in Fig.1c nicely illustrates the developmental differences among the different stages and also the differences between leaf, mature and premature embryos. An inspection of the TEs that correspond to the developmentally dynamic 24-nt siRNAs identified two main classes of TEs (Class A and B) showing clear differences with regard to TE length and more importantly chromatin state. The authors also generated methylomes from preglobular, early heart, bent-cotyledon embryos, leaves, and floral buds. They discovered a clear association of 24-nt siRNAs with differentially methylated euchromatic and heterochromatic TEs during embryogenesis. In addition, by profiling the linker histone 1 h1 mutant, the authors found that the chromatin state alone can explain 24-nt siRNA production from TEs. The presented manuscript is clearly written and well-organized. The experiments/data analyses are logical and well-described and the findings should be of interest to a broad audience. Overall the manuscript is in good shape and should be considered for publication in Genome Biology. I have only a couple of suggestions for improvement.

Thank you for the positive comments. We are glad that you appreciate our manuscript.

1. *It would be nice to see some of the developmental DMRs in a genome browser screenshot to get a better idea of the differences between the profiled developmental stages. It also helps to assess data quality. Ideally, the authors should provide a genome browser link that includes the newly generated datasets as well as the used publicly available datasets for the reviewers so that the quality of their data can be directly inspected.*

We agree that displaying developmental DMRs in a genome browser would help readers examine differences between developmental stages, and have included developmental DMRs in Additional file 2: Figure S4 of the revised manuscript. To allow direct inspection of the full datasets, we deposited processed BigWig/BED files of the methylome and sRNA-seq datasets generated in this study to both NCBI GEO (GSE152971) and a publicly available GitHub page (<https://github.com/mnodine/Papareddy.2020>). Moreover, we have provided summary statistics for libraries that were publicly available in Additional file 1: Table S1 to help assess data quality.

2. *The authors use the term chromatin state in a traditional way (euchromatin and heterochromatin) which is correct but is in my opinion a bit confusing since many readers including myself associate chromatin states with a more detailed classification based on various chromatin features (histone mods, etc.). The authors should clarify that.*

Thank you for the suggestion. To provide a more detailed classification of chromatin states of heterochromatic and euchromatic transposons, we have added metaplots of various chromatin features (i.e. GC content, nucleosome occupancy, and H3K9me2 and H1.1/H1.2 levels) to Additional file 2: Figure S2G of the revised manuscript.

3. *The title needs to be improved. It sounds too general and should better reflect the findings*

We have changed the title from “Chromatin Regulates Bipartite-Classified Small RNA Expression to Maintain Epigenome Homeostasis in Arabidopsis” to “Chromatin Regulates Expression of Small RNAs to Help Maintain Transposon Methylome Homeostasis in Arabidopsis” to better reflect the main findings of our study that chromatin can regulate siRNA expression throughout plant development which in turn can help restore transposon methylation levels.

4. *Little issue with some references on page 14,
" Based on available sRNA-seq datasets, euchromatic, but not heterochromatic, TE-derived 24-nt siRNAs were substantially reduced in mutants deficient in CG and CHH methylation (Fig. 6b and Fig. S6a) (Law et al.,2013; Lister et al., 2008; Stroud et al., 2014)."*

Thank you for noticing our typographical error. We have reformatted the citations properly in the revised manuscript.

Reviewer #2

The manuscript titled "Chromatin Regulates Bipartite-Classified Small RNA Expression to Maintain Epigenome Homeostasis in Arabidopsis" analyzed the dynamics of siRNAs during embryo development in Arabidopsis. The authors found that siRNA expression can be clustered into two main populations, one from TEs in euchromatic regions and the other from TEs in heterochromatic (peri-centromeric) regions. By comparing the difference in siRNA levels from the two types of TEs and correlate that with DNA methylation levels, the authors argue that the compactness of chromatin is a main determinant of the siRNA levels during embryo development. They tested this hypothesis in vegetative tissue using mutants defective in chromatin compaction or specific epigenetic modifications and found that loss of the linker histone H1 results in elevated siRNA levels from TEs in peri-centromeric regions. They thus proposed that chromatin decondensation facilitates Pol IV activity during early embryogenesis, which functions to promote heterochromatin homeostasis and TE silencing. Overall, I think the main conclusions are supported by the data. The writing and data presentation are clear. Importantly, this work is one of the few that study the epigenomic changes during embryo development; the view of cell autonomous regulation of heterochromatin during embryo development should be of interest to a wide range of plant biologists.

We appreciate your positive remarks and are glad that the message we were trying to convey was clear to you.

With that said, I do have a few minor suggestions:

- 1. Methods section should include more details considering that data analysis a main part of this paper. The current description of sRNA and DNA methylome analyses are too simple; how data from this study are compared to the published data should be included.*

We agree that the Methods section should have sufficient details for the readers, and have included more extensive details in the "Small RNA Profiling", "Small RNA Sequencing Analysis", "DNA Methylation Profiling" and "DNA Methylation Analysis" subsections of the Methods (Pages 18-21) in the revised manuscript. In addition, we have uploaded BigWig/BED files of the methylome and sRNA-seq datasets generated in this study to NCBI GEO (GSE152971), and BigWig/BED files for all datasets analyzed to a publicly available GitHub page (<https://github.com/mnodine/Papareddy.2020>). These files can be downloaded by readers and used to compare datasets side-by-side in a genome browser (e.g. the Integrated Genomics Viewer). We have also provided details about the publicly available data analyzed in our study to Additional file 1: Table S1 to help compare the quality of datasets analyzed in our study.

- 2. Page 10, 1st paragraph, Line 4-6. I don't think figure 4d represents long heterochromatic TEs.*

Thank you for pointing this out. We did not include that Figure 4d represents long heterochromatic TEs in the corresponding legend. Sorry for our mistake. We have also

color-coded the graphs in Figure 4 as was done in the rest of the paper to help orientate readers.

3. *Page 10, 2nd paragraph, last 3 lines. I don't quite understand what the sentence means.*

We agree that this sentence was confusing. In the revised manuscript (middle of page 10), we shortened this sentence to focus on siRNA production from both the bodies and edges of long heterochromatic TEs. This was the main point of this paragraph and we hope this improves the clarity.

4. *Page 12, last paragraph, line 3. The reference by Yang DL et al. 2016 should also be included for the characterization of siRNA precursors.*

We have fixed our mistake in the revised manuscript by including the Yang DL et al. reference who also reported that Pol IV-dependent siRNAs accumulate in *dcl2/3/4* mutants.

5. *Abbreviation of RNA polymerase IV should be "Pol IV" instead of "PolIV".*

We have corrected the abbreviation for RNA polymerase IV in the revised manuscript.

6. *GEO accession number should be provided to the reviewers.*

We have deposited the raw and processed next-generation sequencing datasets used in this study to NCBI GEO (GSE152971). This data is now publicly available.